



# C-Coupler3.0: an integrated coupler infrastructure for Earth system modeling

Li Liu[1], Chao Sun[1], Xinzhu Yu[1], Hao Yu[1], Qingu Jiang[1,2], Xingliang Li[2], Ruizhe Li[1], Bin Wang[1,3], Xueshun Shen[2], Guangwen Yang[4,1]

[1]Ministry of Education Key Laboratory for Earth System Modeling, Department of Earth System Science, Tsinghua University, Beijing, China

[2]CMA Earth System Modeling and Prediction Center, China Meteorological Administration, Beijing 100081, China

[3]State Key Laboratory of Numerical Modeling for Atmospheric Sciences and Geophysical Fluid Dynamics (LASG), Institute of Atmospheric Physics, Chinese Academy of Sciences, Beijing, China

[4]Department of Computer Science and technology, Tsinghua University, Beijing, China

*Correspondence to:* L. Liu (liuli-cess@tsinghua.edu.cn)

**Abstract.** The Community Coupler (C-Coupler) for Earth system modeling is a coupler family that has been developed in China since 2010. C-Coupler3.0, the latest version, is fully compatible with the previous version, C-Coupler2, and is an integrated infrastructure with new features. i.e., a series of parallel optimization technologies for accelerating coupling initialization and reducing memory usage, a common halo-exchange library for developing a parallel version of a model, a common module-integration framework for integrating a software module (e.g., a flux algorithm, a parameterization scheme, and a data assimilation method), a common framework for conveniently developing a weakly coupled ensemble data assimilation system, and a common framework for flexibly inputting and outputting fields in parallel. Specifically, C-Coupler3.0 is able to handle coupling under much finer resolutions (e.g., more than 100 million horizontal grid cells) with fast coupling initialization and successful generation of remapping weight files.

## 1 Introduction

Couplers as well as coupling software (Hill et al., 2004; Larson, et al., 2005; Balaji et al., 2006; Craig et al., 2005, 20012, 2017; Redler et al., 2010; Liu et al., 2014, 2018; Hanke et al., 2016) have been widely used in building the coupled models for weather forecasting and climate simulation/prediction. The Community Coupler (C-Coupler) family, whose development was initiated in 2010, has already been widely used for the model development in China (Zhao et al., 2017; Lin et al., 2020; Li et al., 2020; Ren et al., 2021; Shi et al., 2021). The first generation, C-Coupler1 (Liu et al., 2014), which was finished in 2014, can handle the data transfer between different models and the data interpolation (including 2-D and 3-D) between different grids. C-Coupler2 (Liu et al., 2018) released in 2018 further improves the flexibility and user-friendliness with a set of new features, e.g., coupling configurations with new application programming interfaces (APIs) and XML





configuration files, automatic coupling procedure generation, dynamic 3-D coupling under time-evolving vertical coordinate values, etc. To further help the model development in China, the next generation C-Coupler3.0 becomes an integrated infrastructure with a series of new features:

1) C-Coupler3.0 can handle coupling under much finer resolutions (e.g., with more than 100 million horizontal grid cells) with fast coupling initialization. This is achieved through a series of parallel optimization technologies; i.e., parallel triangulation of a horizontal grid (Yang et al., 2019), a distributed implementation of routing network generation for data transfer (Yu et al., 2020), distributed management of horizontal grids and the corresponding parallel remapping weights, and parallel input/output (I/O) of remapping weights from/to a file.

2) C-Coupler3.0 can help to conveniently parallelize a component model with a common halo-exchange library. This library can support various horizontal grids and halo regions and can simultaneously handle the halo exchange for multiple fields for better parallel performance of the model.

3) A new framework in C-Coupler3.0 (hereafter called the common module-integration framework) enables a model to conveniently integrate and then use a software module (e.g., a flux algorithm, a parameterization scheme, or a data assimilation method). This framework can automatically and efficiently handle argument passing between a model and a module, even when they use different data structures, grids, or parallel decompositions.

4) C-Coupler3.0 includes a common framework (Sun et al., 2021) for conveniently developing a weakly coupled ensemble data assimilation (DA) system. This framework provides online data exchanges between a model ensemble and a DA method, for better parallel performance.

5) C-Coupler3.0 includes a common framework (Yu et al., 2022) for flexible parallel inputting and outputting of fields. This framework has already been used by C-Coupler3.0 for improving the input and output of restart fields, and it may also benefit models.

The development of C-Coupler3.0 is based fully on the prior version, so the new coupler is fully compatible with C-Coupler2; i.e., model developers can easily upgrade a coupled model from C-Coupler2 to C-Coupler3.0 without modifying model codes or existing XML configuration files.

The remainder of this paper is organized as follows. Sections 2 to 6 respectively introduce the parallel optimization technologies, halo-exchange library, module-integration framework, data assimilation framework and data input/output framework. Section 7 empirically evaluates C-Coupler3.0, and conclusions and a discussion are provided in Section 8.

## 2   New parallel optimization technologies

Ever finer resolution is a perennial object of model development. Any new coupler should be developed to support finer-resolution coupling. However, we developed C-Coupler2 with the main focus of making it as flexible and user friendly as possible. As a result, C-Coupler2 introduced high overheads in coupling initialization; e.g., it takes more than 80 s to initialize 2-D coupling for about one million horizontal grid points (representing a resolution of ~25 km at a global scale)





using one thousand processor cores (cf. figure 8 in Liu et al., 2018). This becomes a bottleneck in the application of coupled models, and several main shortcomings have emerged:

1) The global management of model grids is very memory consuming. C-Coupler2 keeps the data of each model grid in memory for remapping weight generations. When C-Coupler2 outputs fields into files, grid data will also be outputted. Global management that means that each process keeps the data of all cells (or points) of a grid will use excessive memory at very fine model resolutions. For example, a global model at 3 km resolution will have more than 50 million cells in a horizontal grid, and thus the global management of such a grid will take more than 4 GB memory (given that each grid cell has one center point and four vertexes and C-Coupler uses double-precision floating points in grid management).

2) The sequential triangulation of horizontal grids for generating remapping weights requires much time and memory. C-Coupler2 can automatically generate remapping weights between two different horizontal grids. This requires the vertexes of each grid cell; they can be user specified, or C-Coupler2 can generate them automatically by triangulation to improve user friendliness.

3) A global implementation of routing network generation for data transfer is time consuming. C-Coupler2 can transfer fields between different component models or within one component model. C-Coupler achieves data transfer through $M \times N$ communication (Jacob et al., 2005) following a routing network. Routing networks are generated when initializing model coupling.

4) The global implementation of obtaining remapping weights from a file also requires much time and memory. C-Coupler2 can use the remapping weights from an existing file for the data interpolation between two horizontal grids. During global implementation, each process reads in all weights from a file but uses only part of the weights for parallel data interpolation.

To address the above shortcomings, we should develop distributed management of horizontal grids, parallel triangulation of a horizontal grid, distributed implementation of routing network generation, parallel calculation of remapping weights, and parallel input of remapping weights from a file.

## 2.1 Distributed management of horizontal grids

As it is impractical for each process to keep all the data of a very fine horizontal grid, the data should be kept and managed in a distributed manner; i.e., each process keeps the data of a subset of grid cells, while all processes work cooperatively for horizontal grid management. To investigate how to divide a horizontal grid into subsets of grid cells, C-Coupler2's use of horizontal grids should be studied. There are four types of usage:

1) C-Coupler2 outputs a horizontal grid to a data file with model fields.

2) Two horizontal grids are compared to determine the necessity of data interpolation; e.g., data interpolation will be performed when the two grids are not exactly the same.





3) Two horizontal grids are used for generating remapping weights online.

4) C-Coupler2 enables a process to get the data of a specific subset of grid cells. The whole domain corresponding to a horizontal grid is generally decomposed for parallel integration among processes, where each process conducts integration for a number of local grid cells (such decomposition is called parallel decomposition hereafter). A process may want to get the data of its local grid cells.

Corresponding to the above usage, we designed three types of horizontal grid cell distribution among processes: index-based distribution, domain-based distribution, and parallel-decomposition-based distribution. Transformation between different types of grid distribution was also developed.

### 2.1.1 Index-based distribution

In C-Coupler2, each grid is associated to a component model that runs on a set of processes. Index-based distribution evenly distributes cells of a horizontal grid among all processes of the corresponding component model according to the global indexes of the cells (e.g., each cell has a unique global index between 1 and the grid size), where each process keeps a portion of cells with successive global indexes. For example, given a component model running on four processes with a horizontal grid of 10,000 cells, the four processes keep cells numbered 1–2500, 2501–5000, 5001–7500, and 7501–10,000, respectively.

Index-based distribution enables all processes of a component model to directly use parallel I/O supports (e.g., MPI-IO and PnetCDF) to output a horizontal grid. Moreover, it enables a component model to compare two horizontal grids in parallel; i.e., each process can perform the comparison for only a portion of the grid cells it keeps.

### 2.1.2 Domain-based distribution

A fundamental operation of generating remapping weights is searching the source grid cells around a destination grid cell. In general, these source grid cells are located within a circle centered on the destination grid cell. Index-based distribution is not suitable for searching adjacent source grid cells, as it does not consider the location of each grid point. Therefore, we designed a domain-based distribution that divides a grid domain into a number of subdomains distributed to different processes. Given two horizontal grids, the domain-based distribution of each grid is constructed as follows.

1) Calculate the union domain of the two grid domains.

2) Divide the union domain into a number of non-overlapping subdomains, each of which is assigned to a process.

3) For each grid cell, confirm the subdomain it belongs to and then distribute it to the corresponding process.

Specifically, a regular division of a grid domain based on the longitude and latitude coordinates is employed; i.e., the whole domain of a grid is represented as a rectangle, each polar subdomain is a circle with the boundary of a uniform latitude value, and all non-polar subdomains are rectangles with the same length and width. All subdomains are labeled with





successive IDs and are assigned to a set of processes in a round-robin manner. Such an implementation can easily confirm the subdomain corresponding to the coordinates of a grid cell, the subdomains around a subdomain, and the owner process corresponding to a subdomain ID.

### 2.1.3    Parallel-decomposition based distribution

5        The parallel-decomposition-based distribution in C-Coupler3.0 is not new, as it was implemented in both prior versions. When registering a parallel decomposition on a horizontal grid to C-Coupler, a grid distribution corresponding to the parallel decomposition is generated, where each process of the corresponding component model keeps only the local cells determined by the parallel decomposition.

### 2.1.4    Transformation among grid distributions

10      Different distribution types of the same horizontal grid generally co-exist in a coupled model integration. Distributed grid management should guarantee consistency among the different distribution types of the same grid; i.e., the same grid cell having the same data values throughout different types of distribution. To achieve such consistency, we refer to the index-based distribution as the primary distribution. When registering a horizontal grid to C-Coupler, its index-based distribution is generated first, and the domain- or parallel-decomposition-based distribution is generated from the index-based distribution when required. Such generation will lead to rearrangement of grid data among a set of processes. To achieve parallel rearrangement without collective communications, the existing data transfer functionality of C-Coupler is employed to generate a domain- or parallel-decomposition-based distribution from the index-based distribution.

## 2.2      Parallel triangulation of a horizontal grid

To achieve parallel triangulation, we upgraded the sequential procedure in the previous C-Couplers into a parallel procedure. Specifically, this upgrade employs a parallel algorithm, PatCC1 (**Pa**rallel **t**riangulation algorithm with **C**ommonality and parallel **C**onsistency, version **1**), based on Delaunay triangulation (Su and Drysdale, 1997). PatCC1 performs three major steps for triangulating a horizontal grid (see Yang et al. (2019) for further details): 1) it divides a global grid (each process keeps the data of all grid cells) into a number of non-overlapping kernel subgrid domains; 2) each kernel subgrid domain is properly enlarged into an expanded subgrid domain; and 3) each process independently conducts triangulation for a subset of expanded subgrid domains. PatCC1 then checks whether the triangulation results are consistent among the expanded subgrid domains; i.e., if the kernel subgrid domains of two expanded subgrid domains have a common boundary, the triangulation results from these two expanded subgrid domains must be the same on the common boundary.





Adapting PatCC1 to C-Coupler3.0 requires modifications. As C-Coupler3.0 keeps a horizontal grid in a distributed manner rather than globally, the domain-based distribution is generated instead of following the first major step of PatCC1; e.g., each subdomain is exactly a kernel subgrid domain of PatCC1. Moreover, each kernel subgrid domain is also enlarged based on the domain-based distribution.

In theory, the vertexes of the grid cells should originate from the Voronoi diagram generated from the triangulation results; i.e., each vertex should be the circumcenter of a triangle. However, C-Coupler1 and C-Coupler2 implement a simple, imprecise approximation, whereby the point at the average longitude and latitude values of the three vertexes of a triangle is used as the circumcenter. To improve vertex generation, C-Coupler3.0 uses the real circumcenters after removing irregular triangles. An irregular triangle has its circumcenter outside itself. The vertexes of grid cells are also generated in parallel; i.e.,

each process only generates vertexes within its kernel subgrid domains.

### 2.3    Parallel remapping weights calculation

Both C-Coupler1 and C-Coupler2 can generate the remapping weights from a source horizontal grid to a destination horizontal grid online. C-Coupler2 achieves parallel calculation of remapping weights based on the global management of model grids, where a process only works for the local grid cells on a parallel decomposition of the destination grid. This

parallel implementation is further improved in C-Coupler3.0 to accommodate the distributed management of horizontal grids. Specifically, the remapping weights are calculated under the domain-based distribution, and different processes work for different grid subdomains. When a process calculates remapping weights for a subdomain, it can keep only the destination grid cells in this subdomain, whereas it should have the source grid cells in a larger subdomain (referred to hereafter as the source subdomain). It is possible that the current source subdomain does not contain all source grid cells required to

calculate the remapping weights for a destination grid cell (e.g., especially when extrapolation is enabled), so the source subdomain should be further enlarged by obtaining the corresponding source grid cells from the corresponding processes. An implementation based on MPI one-sided communication functionality has been developed for enlarging a source subdomain.

### 2.4    Parallel input of remapping weights

Instead of generating the remapping weights online, C-Coupler2 can use offline remapping weights from a file when

this file matches the corresponding source and destination horizontal grids. To reduce memory consumption and expedite operation, C-Coupler3.0 achieves parallel input of remapping weights as follows.

1) Processes cooperatively read in remapping weights in parallel from a file; i.e., each process reads in a subset of the remapping weights.

2) Processes cooperatively and in parallel sort remapping weights in ascending order of destination cell index (a remapping

weight generally comprises the source cell index, destination cell index, and weight value).





3)   Processes cooperatively rearrange the sorted remapping weights to make each process obtain the subset of remapping weights it requires locally.

## 2.5      Distributed routing network generation

C-Coupler1 and C-Coupler2 employ the data transfer functionality only for model coupling invoked by API calls from component models. However, C-Coupler3.0 uses this functionality more frequently internally; e.g., for transformation among grid distributions, halo exchange (Section 3), passing an argument between a model and a software module (Section 4), online data exchanges between a model ensemble and a DA method (Section 5), and data rearrangement for parallel I/O (Section 6). Although C-Coupler1 and C-Coupler2 use an $M \times N$ communication approach (Jacob et al., 2005) that transfers data with parallel communications, data transfer is initialized using a global implementation of routing network generation that relies on inefficient gather–broadcast communications. To accelerate routing network generation, a distributed implementation of routing network generation (DiRong1.0) was designed and implemented in C-Coupler3.0 (see Yu et al. (2020) details). DiRong1.0 does not introduce any gather–broadcast communication via distributed sorting of the corresponding routing information tables.

## 3    Common halo-exchange library

The development of a component model generally includes parallelizing a serial program to allow effective use of a large number of processor cores to accelerate model integration. A typical approach to model parallelization is to decompose the whole grid domain into a number of subdomains, assign each subdomain to a process, and then make different processes conduct the integration for different subdomains. To achieve correct parallelization that maintains almost exactly the same simulation results as the original serial program, processes in a parallel execution should work cooperatively; e.g., processes should exchange data among themselves when required.

Given a component model running on four processes, Fig. 1 shows a parallel decomposition of the horizontal grid, where each subdomain is assigned to each process. As data in grid cells are generally interdependent (i.e., the integration calculation on one grid cell depends on the results for the grid cells around it), correct parallelization requires enlarging a process's subdomain using a certain halo region to keep the results from other processes. This requires processes to cooperatively exchange data to gather the results for these halo regions (called halo exchange for short). Corresponding to Fig. 1, Fig. 2 shows the halo region for each process and the source processes of the results in each halo region.

Implementing halo exchange is a fundamental part of developing model parallelization. Various component models have their own halo-exchange libraries or procedures based on MPI communications (Dennis et al., 2012; Wang et al., 2014; Deconinck et al., 2017; Adams et al., 2019). To aid the parallelization of various component models in China, it would be beneficial to develop a common halo-exchange library that can work for any model grid, parallel decomposition and halo



region. A fundamental part of halo exchange is the transfer of data among the processes of a component model. As C-Coupler2 can handle data transfer within a component model, we propose to use it as the foundation for developing a common halo-exchange library. Moreover, C-Coupler2 provides common representations for 1-D to 4-D grids and parallel decompositions. Its API *CCPL_register_normal_parallel_decomp* (API 1 in the Appendix) for registering a parallel

decomposition on a grid allows each grid cell to be specified in the local subdomains of each process. Specifically, a cell calculated by the current process can be specified using its global index, whereas a grid cell in a halo region can be labeled using a special value (i.e., *CCPL_NULL_INT*).

To support halo exchanges in C-Coupler3.0, a new API *CCPL_register_halo_region* (API 2 in the Appendix) is designed to describe a halo region based on the corresponding parallel decomposition. This enables specification of the

global index of each grid cell in the halo region. Users can further register a halo exchange handler for a number of fields via the API *CCPL_register_halo_exchange_interface* (API 3 in the Appendix) and then conduct halo exchange via the API *CCPL_execute_halo_exchange_using_id* (API 4 in the Appendix). The API *CCPL_finish_halo_exchange_using_id* (API 5 in the Appendix) can be used to further achieve asynchronous communications. Halo exchange is performed by C-Coupler3.0 internal import/export interfaces using data transfer operations that are automatically generated when registering

a halo exchange handler.

The above implementation based on C-Coupler2 achieves the following advantages.

1) Commonality. Besides the common representations for grids and parallel decompositions, the representation of halo regions is also common, as each grid cell in a halo region is specified independently. As a result, the halo-exchange library allows a component model not to conduct exchanges for some grid cells (e.g., land cells in an ocean model) for

better parallel performance.

2) Convenience. The halo-exchange library only requires users to describe the halo region, the corresponding parallel decomposition, and the fields to be exchanged, whereas the parallelization operations (i.e., generation of a routing network among processes, MPI communications following the routing network, and packing/unpacking of data values) are automated.

3) Efficiency. The halo-exchange library allows multiple fields to be bundled together in MPI communications for better parallel performance.

The halo-exchange library further provides two new APIs, *CCPL_report_one_field_instance_checksum* and *CCPL_report_all_field_instances_checksum* (APIs 6 and 7 in the Appendix), which calculate and then report the global checksums of fields that have been registered to C-Coupler3.0. In most cases, correct parallelization of a code segment

should not change the global checksum of each model field; i.e., the global checksum of each field should remain the same under different parallel settings (e.g., serial or parallel run, different numbers of processes, or different parallel decompositions). These two APIs can conveniently check whether the parallelization of a code segment is correct and thus enable users to develop a parallel version of a model incrementally in a segment-by-segment manner.





## 4  Common module-integration framework

The development of a model always requires integration of a new software module into the model; for example, integrating a parameterization scheme into the physical package of an atmosphere model. Such integration is traditionally achieved internally (referred to hereafter as internal integration), whereby the new module becomes a native procedure that is directly called by the model. Internal integration generally requires the source code of the new module to be adapted to the compilation system and data structures of the model, which can introduce much work in some cases. For example, model developers should compile and statically link together codes of the model and the new module, while possibly having to change the names of some variables if the model and the new module apply the same name to different common variables. Data dependencies in the horizontal direction are always neglected in existing physical parameterization schemes for atmosphere models when the horizontal grid interval is large, as has been the case in the past. As a result, single-column data structures have been used in the physical packages of atmosphere models for better data access locality. To improve the simulation results under finer horizontal resolutions, some 3-D parameterization schemes that consider data dependencies in the horizontal direction are being developed (Zhang et al., 2018; Veerman et al., 2019). As a 3-D parameterization scheme cannot use the single-column data structures for implementation and its parallelization may rely on halo exchanges in the horizontal direction, there will be many technical challenges when integrating a 3-D parameterization scheme into a single-column physical package.

External integration is another methodology for integration. A model indirectly calls a module based on an integration framework; i.e., the model calls the APIs of the framework, then the framework calls the module. External integration offers the following advantages over internal integration.

1) Independence. The module can remain independent of the model; i.e., the module does not become a native procedure of the model and thus can have its own compilation system or even its own data structures, parallel decompositions, and grids.

2) Convenience. When there are inconsistencies (e.g., in terms of data structure, parallel decomposition, or grid) between the module and the model, the framework can automatically handle them without introducing extra work to users.

A framework should be able to pass arguments between the model and the module. The Common Community Physics Package (CCPP; Heinzeller et al, 2022) achieves this through memory sharing; i.e., each field in the argument list of the module is essentially a common variable of the model. CCPP, therefore, requires the model and the module to use the same data structure, parallel decomposition, and grid and thus does not fully achieve the above advantages. To develop a common module-integration framework, we propose employing C-Coupler for passing arguments because it can automatically handle the coupling between different grids or different parallel decompositions.

A framework should be able to call the modules with various argument lists, whereas a programming language such as Fortran or C/C++ requires a caller to match the specific argument list of a callee. It is impractical to make the codes of a framework enumerate the callers corresponding to all possible argument lists. To overcome this challenge, CCPP designs a





text rule to describe the information of all arguments of a module and thus can follow the rule to automatically generate the caller's code for calling the module. We do not recommend this because it forces users to study the new rule and it is difficult to design a common rule that can describe an argument with any grid and any parallel decomposition. We prefer a new solution of designing a rule for writing driving procedures that can flexibly declare each argument of the module via the APIs of the framework.

With the experiences learned from CCPP, we prefer to use a dynamic-link library (DLL) to integrate a module. This technique guarantees independence between the model and the module and maintains the privacy of the module; i.e., module developers need only provide the binary of DLL without the source code in a community development of the model.

Overall, we designed the architecture of the common module-integration framework based on C-Coupler. This

framework consists of an external module-integration manager, an argument coupling manager, an external module driving engine, and hybrid interfaces with APIs and a configuration file (see sections 2.12 and 3.7 in the C-Coupler3.0 User Guide for further details). The following subsections further examine the detailed implementation of the integration framework.

## 4.1    External module-integration manager

The external module-integration manager employs the DLL technique; i.e., an external module is compiled into a DLL

that is loaded when the model launches the module. To integrate an external module, users should develop the driving procedures; i.e., an initialization driving subroutine, an execution driving subroutine, and a finalization driving subroutine. The initialization driving subroutine is used for declaring the arguments and driving the initialization of the external module; it only has one input argument, an ID for labeling the current external module (called the module ID). The execution driving subroutine is used to drive the execution of the external module. It has two input arguments, the module ID and a chunk

index that can be used for single-column parameterization schemes. The finalization driving subroutine is used to drive the finalization of the external module and does not have any argument.

Each external module has a unique name that determines the subroutine name of each driving procedure. For example, given an external module named "EM", its three driving procedures should be named "EM_*CCPL_INIT*",

"EM_*CCPL_RUN*", and "EM_*CCPL_FINALIZE*", respectively. A DLL can include multiple external modules, and each module should have a unique name. The module name and the corresponding DLL are specified in the configuration file when users want to use the corresponding external module. As a result, users can flexibly change the external module used by modifying the configuration file without modifying the model code.



## 4.2 Argument coupling manager

Each argument that an external module passes with a model should be declared in the initialization driving subroutine via the new API *CCPL_external_modules_declare_argument* (API 8 in the Appendix). The corresponding grid and parallel decomposition should be specified when declaring a module's argument. An external module can have its own grid or

parallel decomposition but can also share the same grid and the same parallel decomposition with the model; e.g., the new APIs *CCPL_external_modules_para_get_field_grid_ID* and *CCPL_external_modules_para_get_field_decomp_ID* (APIs 9 and 10 in the Appendix) enable the external module to obtain the model's grid and parallel decomposition, respectively. The memory space of a module's argument can be allocated either explicitly by the external module or implicitly by the common module-integration framework. Moreover, a module's argument can share the memory space of the corresponding model

field when it uses the same grid and parallel decomposition as the model field.

The argument coupling manager is responsible for exchanging values between each pair of a module's argument and the corresponding model field; i.e., each input argument obtains values from the corresponding model field before calling the execution driving subroutine, then each output argument returns values to the corresponding model field when the execution driving subroutine finishes. The values can be exchanged by the coupling functionality of C-Coupler, which enables an input

(or output) argument and the corresponding model field to use different memory space, grids, parallel decompositions, or data structures.

## 4.3 External module driving engine

The external module driving engine enables a model to initialize, execute, and finalize a set of external modules that have been enclosed in DLLs through calling the APIs *CCPL_external_modules_inst_init*, *CCPL_external_modules_inst_run*,

and *CCPL_external_modules_inst_finalize* (APIs 11–13 in the Appendix). The API *CCPL_external_modules_inst_init* is responsible for creating an instance of the external modules. For each external module, the corresponding DLL is loaded when required, and then the initialization driving subroutine is called to initialize the external module. After returning from the initialization driving subroutine, the common module-integration framework knows all input and output arguments of the external module, and then its argument coupling manager generates two coupling procedures: an input coupling procedure

for passing the values of the input arguments from the model to the external module, and an output coupling procedure for passing the values of the output arguments from the external module to the model. A set of model fields should be specified when calling the API *CCPL_external_modules_inst_init* (corresponding to the parameter "*field_inst_ids*"). For each argument of the external module, there must be a corresponding model field; i.e., the argument and the model field have the same name, or the mapping between their different names has already been specified in the corresponding configuration file.

The API *CCPL_external_modules_inst_run* is responsible for executing an external module. It first executes the input coupling procedure to make the external module obtain the input values from the model. Next, it calls the corresponding execution driving subroutine to run the external module. Finally, it executes the output coupling procedure to make the



model obtain the return values from the external module. The API *CCPL_external_modules_inst_finalize* calls the corresponding finalization driving subroutine to finalize the external module.

## 5    Common ensemble coupled data assimilation framework

5    DAFCC1 (Sun et al., 2021), a DA framework based on C-Coupler2.0, is now part of C-Coupler3.0, and it enables users to conveniently develop a weakly coupled ensemble DA system. It benefits greatly from other functionalities of C-Coupler3.0. For example, its initialization has been accelerated by new parallel optimization technologies; the integration of a DA method and online data exchanges between a model ensemble and a DA method are achieved by the common module-integration framework. The C-Coupler3.0 User Guide and Sun et al. (2021) give further details of DAFCC1.

## 10    6    Common data input/output framework

CIOFC1, a common data I/O framework based on C-Coupler2.0, is now part of C-Coupler3.0. CIOFC1 can adaptively input data fields from a time-series dataset during model integration; interpolate data in 2-D, 3-D, or the time dimension automatically when necessary; and output fields either periodically or irregularly. It helps C-Coupler3.0 to input and output restart fields in parallel and further provides APIs and configuration files to enable a component model to conveniently use parallel I/O and enable users to flexibly specify I/O settings; e.g., the model fields for I/O, the time series of the data files for I/O, and the data grids in the files. The C-Coupler3.0 User Guide and Yu et al. (2022) give further details of CIOFC1.

## 7    Empirical evaluation

This section reports empirical evaluation of C-Coupler3.0, including the parallel optimization technologies, common halo-exchange library, and common module-integration framework. For the evaluation of the DA framework DAFCC1 and the I/O framework CIOFC1, please refer to Sun et al. (2021) and Yu et al. (2022), respectively.

All test cases in this section were run on the High-Performance Computing System (HPCS) of the Earth system numerical simulator (http://earthlab.iap.ac.cn/en/), which has 70 PB parallel storage capacity and ~100,000 CPU cores running at 2.0 GHz. Each computing node includes 64 cores and 256 GB memory. All nodes are connected by a network system with a communication bandwidth up to 100 Gbps. All codes of the test coupled model are compiled by an Intel Fortran and C++ compiler (version 17.0.5) at the O2 optimization level using an OpenMPI library.





## 7.1 Evaluation of the parallel optimization technologies

To evaluate the impact of the parallel optimization technologies, we developed a test coupled model that consists of two toy component models with two-way coupling based on C-Coupler3.0. The test coupled model allows us to flexibly change the model settings in terms of grid size and number of processor cores (processes).

We first evaluate the coupling initialization (using offline remapping weight files) in the test coupled model in terms of time and memory usage. The results in Table 1 demonstrate that C-Coupler3.0 can initialize coupling quickly for ultra-large grid sizes and many processes (cores). In some cases, faster coupling initialization can be achieved when the component models use more processes. This is because most computation in coupling initialization has been parallelized. As the parallel reading of a remapping weight file cannot be accelerated continuously by increasing the I/O processes, and the global communications in automatic coupling procedure generation always introduce higher overheads using more processes, coupling initialization can become slower by employing more processes of the component models.

Table 1 also shows that the average memory usage of the processes of the test coupled model is affordable (each process/core has an average memory capacity of 4 GB), even when the grid size is extremely large, which demonstrates the effectiveness of the distributed management of horizontal grids in decreasing the memory usage of C-Coupler. Moreover, the parallel optimization technologies appear to enable C-Coupler3.0 to generate large remapping weight files (a bilinear remapping algorithm is used), and the generation can become faster when using more processes.

Based on the coupled models with C-Coupler2, the correctness of the parallel optimization technologies has been verified under a bitwise-identical criterion with the following properties.

1) C-Coupler3.0 achieves bitwise-identical results to C-Coupler2 when using the same offline remapping weight files (some bugs in remapping weight generation have been fixed during the development of C-Coupler3.0).

2) C-Coupler3.0 achieves bitwise-identical results (including the remapping weight file and the fields after coupling) using different numbers of processes.

## 7.2 Evaluation of the common halo-exchange library

To evaluate the common halo-exchange library, we developed a test component model that allowed us to flexibly change the grid size, the halo regions, the model fields corresponding to halo exchanges, and the number of processes.

The halo-exchange library allows multiple fields with different dimensions to be bundled in the same halo-exchange operation in order to improve performance. For example, multiple 2-D and 3-D fields can be bundled together. To evaluate the impact of this functionality, we designed different bundle settings corresponding to five 2-D and five 3-D fields. Table 2 shows the performance of halo exchange for different sizes of horizontal grids and different numbers of processes. The results show that field bundling can significantly improve the performance of halo exchange. This is because the performance of data transfer by MPI generally depends on the size of the data transferred, with a smaller data size generally





resulting in lower performance. Although the size of the data transferred by each process in halo exchange is generally small, field bundling can effectively enlarge the data size of an MPI transfer, thus improving the performance of halo exchange.

When parallelizing a model with a rectangle-based parallel decomposition, the halo region is generally regular with a fixed number of levels around each rectangle. For an ocean model, the land cells in each halo region are useless and can be
neglected during halo exchange. The halo-exchange library improves performance by enabling the convenient neglect of useless cells in each halo region. To evaluate the impact of this functionality, we design a test case with a proportion of useless cells distributed randomly in each halo region (four levels). Table 3 shows the corresponding performance of halo exchanges for different sizes of horizontal grid and different numbers of processes when there are five 3-D fields bundled together in each halo exchange operation. The results show that the capability to neglect useless cells can further improve the
performance of halo exchanges, as it can reduce the amount of data transferred among processes.

### 7.3    Evaluation of the common module-integration framework

Sun et al. (2021) confirmed the effectiveness of the module-integration framework, as the DA framework DAFCC1 employs the module-integration framework for integrating the codes of DA methods and for online data exchange between the model and DA methods. To further evaluate the effectiveness, we use the Community Earth System Model (CESM;
Hurrell et al., 2013) version 1.2.1 (called the baseline version) as well as the air–sea flux algorithm used in the model. Specifically, a test version based on the baseline version, where the air–sea flux algorithm is indirectly called by the model via the module-integration framework, was developed as follows.

1) Develop the initialization, execution, and finalization driving subroutines of the air–sea flux algorithm. The initialization driving subroutine makes the air–sea flux algorithm share the same grid, parallel decomposition, and memory space of
each field with the model.

2) Compile the code of the air–sea flux algorithm and the driving subroutines into a DLL.

3) Replace the original call of the air–sea flux algorithm by calling the corresponding APIs of the module-integration framework, and then develop a configuration file corresponding to the air–sea flux algorithm in the DLL.

We find that the test version can achieve bitwise-identical results to the baseline version, thus showing that a flux
calculation algorithm can be correctly integrated into a coupled model using the module-integration framework.

With regard to the performance of the module-integration framework, we developed a test coupled ensemble DA system to evaluate the performance of data exchange in argument passing. This system includes a toy coupled model with two toy component models, where one component model is assimilated with three toy DA algorithms. These DA algorithms are an individual algorithm that operates separately on the data of each ensemble member (called individual DA), an
ensemble algorithm that operates on the ensemble mean data (called ensemble-mean DA), and an ensemble algorithm that operates on the data gathered from all ensemble members (called ensemble-gather DA). The test system allows us to flexibly change the grid size, the number of ensemble members, and the number of processes of the assimilated component model.



Table 4 shows that the data exchange corresponding to the individual DA performs best. As the individual DA operates on the data of each ensemble member separately, different ensemble members can handle the data exchange simultaneously. As an ensemble DA (the ensemble-mean DA or the ensemble-gather DA) operates on the data from the whole model ensemble, the ensemble DA have to exchange data with all ensemble members individually. In this evaluation, we made the

5 ensemble DA run on all processes of the whole model ensemble for maximum parallelism of the ensemble DA, which minimizes the MPI message size in the data exchanges and can result in poor performance of MPI communications. Although the data exchange corresponding to the ensemble DA is much slower than the individual DA, it should be much faster than the I/O on which the DA data exchange in an off-line DA system depends. The results in Table 4 also reveal that the parallel optimization technologies make C-Coupler3.0 initialize a DA algorithm with reasonable overhead. An ensemble

DA algorithm corresponds to a larger initialization overhead than that from an individual DA algorithm. This is because the initialization of an individual DA algorithm only introduces the coupling generation within each ensemble member, whereas the initialization of an ensemble DA algorithm introduces more occurrences of coupling generation (i.e., coupling generation between the whole ensemble and each ensemble member).

## 8   Summary and discussion

As a new version of C-Coupler, C-Coupler3.0 achieves various advancements over its predecessors; e.g., parallel optimizations, halo exchanges for model parallelization, and frameworks for integrating a software module, developing a weakly coupled ensemble data assimilation system, and inputting/outputting fields in parallel. As a result, C-Coupler3.0 is an integrated infrastructure for Earth system modeling and can effectively handle coupling at ultra-high resolutions.

The development of C-Coupler3.0 is based fully on C-Coupler2. All APIs, configuration files, and existing

functionalities of C-Coupler2 are maintained in C-Coupler3.0, making it fully compatible with the prior version, so model developers can easily upgrade a coupled model from C-Coupler2 to C-Coupler3.0 without modifying model codes or existing XML configuration files. As an integrated infrastructure, C-Coupler3.0 provides more functionalities with the same suit of APIs and configuration files, while also keeping flexibility in usage. Model developers can separately use the functionalities for model coupling, halo exchanges, module integration, ensemble data assimilation, and data I/O. C-

Coupler3.0 benefits from the architecture of C-Coupler2, and its code complexity is not increased significantly relative to the prior version. Specifically, C-Coupler3.0 has about 66,000 lines of code and 140 APIs, of which around 30,000 lines of code and 50 APIs are new compared with C-Coupler2.

Our future work will further improve C-Coupler3.0 by, for example, upgrading the DA framework to support a strongly coupled DA and upgrading the data I/O framework to support asynchronous I/O.

*Code availability.* A source code version of C-Coupler3.0 can be viewed via https://doi.org/10.5281/zenodo.7235470 (please contact us for authorization before using C-Coupler for developing a system). The user manual of C-Coupler3.0 is attached



as the supplement. The source code and scripts for the test coupled model used in Section 7.1 can be downloaded from https://doi.org/10.5281/zenodo.7236156. The source code and scripts for the test component model used in Section 7.2 can be downloaded from https://doi.org/10.5281/zenodo.7236138. The source code of the air–sea flux algorithm with C-Coupler3 for software integration (Section 7.3) can be downloaded from https://doi.org/10.5281/zenodo.7237292. The

source code and scripts of the test coupled ensemble DA system used in Section 7.3 can be downloaded from https://doi.org/10.5281/zenodo.7237290.

*Author contributions.* LL led the design and development of C-Coupler3.0 and paper writing. LL, CS, XY, HY, QJ, RL and XL contributed to the code development and validation of C-Coupler3.0. BW, XS and GY contributed to the motivation and

evaluation. All authors contributed to improvement of ideas and paper writing.

*Competing interests.* The authors declare that they have no conflict of interest.

*Acknowledgements.* This work was supported in part by the National Key Research Project of China (grant no.

2017YFC1501903) and jointly supported in part by the Natural Science Foundation of China (grant no. 42075157).

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



# Appendix: introductions to a part of C-Coupler APIs referred in this paper

## API 1: CCPL_register_normal_parallel_decomp

- **integer FUNCTION CCPL_register_normal_parallel_decomp(decomp_name, grid_id, num_local_cells, local_cells_global_index, annotation)**
  - return value [INTEGER; OUT]: The ID of the parallel decomposition.
  - decomp_name [CHARACTER; IN]: The name of the parallel decomposition. It has a maximum length of 80 characters. Each character must be 'A'~'Z', 'a'~'z', 0~9 or '_'.
  - grid_id [INTEGER; IN]: The ID of the corresponding horizontal (H2D) grid that has already been registered to C-Coupler.
  - num_local_cells [INTEGER; IN]: The number of local grid cells ($\geq 0$) in the parallel decomposition of the current MPI process.
  - local_cells_global_index [INTEGER, DIMENSION(:); IN]: The global index of the local grid cells in the parallel decomposition of the current MPI process. The array size of "local_cells_global_index" cannot be smaller than "num_local_cells". Each value in "local_cells_global_index" must be *CCPL_NULL_INT* or a value between 1 and the grid size.
  - annotation [CHARACTER, OPTIONAL; IN]: The annotation used to mark the corresponding model code that calls this API. It has a maximum length of 512 characters.

- **Description of this API**
  This API registers a new parallel decomposition of a horizontal grid ("grid_id") among all MPI processes of the component model corresponding to "grid_id", and returns the ID of the new parallel decomposition when the registration succeeds. The new parallel decomposition corresponds to the same component model with "grid_id". All MPI processes of the component model are required to call this API at the same time, with the same "decomp_name" and the same horizontal grid. This API cannot be called when the coupling configuration stage of the corresponding component model has already been ended.

## API 2: CCPL_register_halo_region

- **integer FUNCTION CCPL_register_halo_region(halo_name, decomp_id, num_local_cells, local_cells_local_indexes, local_cells_global_indexes, annotation)**
  - return value [INTEGER; OUT]: The ID of the new halo region.
  - halo_name [CHARACTER; IN]: The name of the halo region. It has a maximum length of 80 characters. Each character must be 'A'~'Z', 'a'~'z', 0~9 or '_'. There cannot be two halo regions with the same name under the same parallel decomposition.
  - decomp_id [INTEGER; IN]: The ID of an existing parallel decomposition corresponding to the halo region. In Each process, the local sub domains corresponding to the given parallel decomposition should contain all grid cells in the halo regions. When registering a parallel decomposition (Section 2.5), the global index of each cell in halo regions should be set to CCPL_NULL_INT.
  - num_local_cells [INTEGER; IN]: The number of local grid cells ($\geq 0$) in the halo region in the current MPI process.
  - local_cells_local_indexes [INTEGER; DIMENSION(:); IN]: The local index of each grid cell in the halo region of the current MPI process. The local index means the index of a grid cell in the local sub domains of current MPI process corresponding to the given parallel decomposition. The local index starts from 1. The array size of "local_cells_local_indexes" cannot be smaller than "num_local_cells".
  - local_cells_global_indexes [INTEGER; DIMENSION(:); IN]: The global index of each grid cell in the halo region





of the current MPI process. The array size of "local_cells_global_indexes" cannot be smaller than "num_local_cells". Each element of "local_cells_global_indexes" should be between 1 and size of the whole grid domain.

- annotation [CHARACTER, OPTIONAL; IN]: The annotation used to mark the corresponding model code that calls this API. It has a maximum length of 512 characters.

- **Description of this API**

This API registers halo regions on the given parallel decomposition, and returns the ID of the new halo regions when the registration succeeds. All MPI processes of the component model are required to call this API at the same time, with consistent parameters.

## API 3: CCPL_register_halo_exchange_interface

- **integer FUNCTION CCPL_register_halo_exchange_interface(interface_name, num_field_instances, field_instance_IDs, halo_region_IDs, annotation)**
  - return value [INTEGER; OUT]: The ID of the new halo exchange interface.
  - interface_name [CHARACTER; IN]: The name of the halo exchange interface. Each character must be 'A'~'Z', 'a'~'z', 0~9 or '_'. There cannot be two halo exchange interfaces with the same name in the same component model.
  - num_field_instances [INTEGER; IN]: The number of field instances (>0) exchanged via this halo exchange interface.
  - field_instance_IDs [INTEGER, DIMENSION(:); IN]: The ID of the field instances that are exchanged by this interface. All field instances specified by "field_instance_IDs" must correspond to the same component model. The array size of "field_instance_IDs" cannot be smaller than "num_field_instances". Any two field instances cannot share the same field name.
  - halo_region_IDs [INTEGER, DIMENSION(:); IN]: The ID of the halo regions corresponding to the given field instances (each ID corresponds to a field instance). The array size of the parameter "halo_region_IDs" cannot be smaller than the parameter "num_field_instances". All halo regions and all field instances must correspond to the same component model.
  - annotation [CHARACTER, OPTIONAL; IN]: The annotation used to mark the corresponding model code that calls this API. It has a maximum length of 512 characters.
- **Description of this API**

This API registers a new halo exchange interface for a set of field instances, and returns the ID of the new halo exchange interface when the registration succeeds. All MPI processes of the component model are required to call this API at the same time, with consistent parameters.

## API 4: CCPL_execute_halo_exchange_using_id

- **SUBROUTINE CCPL_execute_halo_exchange_using_id(interface_id, is_asynchronous, annotation)**
  - interface_id [INTEGER; IN]: The ID of a halo exchange interface.
  - is_asynchronous [LOGICAL; IN]: A mark of the asynchronous or synchronous mode for executing the interface. When "is_asynchronous" is set to *true*, the interface will be returned immediately without waiting for completement of the halo exchange. When "is_asynchronous" is set to *false*, the interface will not be returned until the halo exchange finishes.
  - annotation [CHARACTER, OPTIONAL; IN]: The annotation used to mark the corresponding model code that calls this API. It has a maximum length of 512 characters.
- **Description of this API**





This API executes a halo exchange interface based on the given interface ID. All MPI processes of the corresponding component model are required to call this API at the same time, with consistent parameters.

**API 5: CCPL_finish_halo_exchange_using_id**

- **SUBROUTINE CPL_finish_halo_exchange_using_id(interface_id, annotation)**
  - interface_id [INTEGER; IN]: The ID of a given halo exchange interface.
  - annotation [CHARACTER, OPTIONAL; IN]: The annotation used to mark the corresponding model code that calls this API. It has a maximum length of 512 characters.
- **Description of this API**
  This API finishes execution of a halo exchange interface based on the given interface ID. All MPI processes of the corresponding component model are required to call this API at the same time, with consistent parameters. If the halo exchange interface is executed asynchronously, this API will wait until the halo exchange finishes; otherwise, this API will return immediately.

**API 6: CCPL_report_one_field_instance_checksum**

- **SUBROUTINE       CCPL_report_one_field_instance_checksum(field_instance_id,       bypass_decomp,       hint, annotation)**
  - field_interface_id [INTEGER; IN]: The ID of a given field instance.
  - bypass_decomp [LOGICAL; IN]: A mark to specify whether to consider the parallel decomposition when calculating the checksum. When its value is "false", the parallel decomposition of the field will be considered.
  - hint [CHARACTER; IN]: The hint used to mark the code position when calling this API. It has a maximum length of 512 characters. It will be outputted when reporting the checksum value.
  - annotation [CHARACTER, OPTIONAL; IN]: The annotation used to mark the corresponding model code that calls this API. It has a maximum length of 512 characters.
- **Description of this API**
  This API calculates and reports the checksum of a field instance throughout the processes of the corresponding component model. The information of checksums will be outputted to C-Coupler's log files under the directory "CCPL_dir/run/CCPL_logs". All MPI processes of the corresponding component model are required to call this API at the same time, with consistent parameters.

**API 7: CCPL_report_all_field_instances_checksum**

- **SUBROUTINE CCPL_report_all_field_instances_checksum (comp_id, bypass_decomp, hint, annotation)**
  - comp_id [INTEGER; IN]: The ID of a the given component model.
  - bypass_decomp [LOGICAL; IN]: A mark to specify whether to consider the parallel decomposition when calculating the checksum. When its value is "false", the parallel decomposition of the field will be considered.
  - hint [CHARACTER; IN]: The hint used to mark the code position when calling this API. It has a maximum length of 512 characters. It will be outputted when reporting the checksum value.
  - annotation [CHARACTER, OPTIONAL; IN]: The annotation used to mark the corresponding model code that calls this API. It has a maximum length of 512 characters.
- **Description of this API**
  This API calculates and reports the checksum of all registered field instances of the given component model throughout the processes of the component model. The information of checksums will be outputted to C-Coupler's log files under the





directory "CCPL_dir/run/CCPL_logs". All MPI processes of the given component model are required to call this API at the same time, with consistent parameters.

**API 8: CCPL_external_modules_declare_argument**

- **INTEGER FUNCTION CCPL_external_modules_declare_argument (proc_inst_id, data_pointer, field_name, type_inout, decomp_id, grid_id, dims_size, field_unit, annotation)**
  - return value [INTEGER; OUT]: The ID of declared argument.
  - proc_inst_id [INTEGER; IN]: The ID of the external module instance.
  - data_pointer [REAL or INTEGER, no DIMENSION or DIMENSION|(:), (:,:), (:,:,:) or (:,:,:,:)|; INOUT]: The data buffer pointer corresponding to the declared argument. The number of dimensions of the data buffer pointer cannot be larger than the array size of "dims_size".
  - field_name [CHARACTER; IN]: The name of the declared argument. It has a maximum length of 80 characters. Each character must be 'A'~'Z', 'a'~'z', 0~9 or '_'. A "field_name" is legal only when there is a corresponding entry in the configuration file "public_field_attribute.xml".
  - type_inout [INTEGER; IN]: The mark for specifying whether the declared argument of the external module instance is an input or output argument. The value of *"CCPL_PARA_TYPE_IN"* means an input argument passed from the component model to the external module. The value of *"CCPL_PARA_TYPE_OUT"* means an output argument passed from the external module to the component model. The value of *"CCPL_PARA_TYPE_ INOUT"* means the declared argument is both input and output.
  - decomp_id [INTEGER; IN]: The ID of the corresponding parallel decomposition. If the declared argument is a scalar variable or a field instance on a grid without a horizontal sub grid (for example, the field instance is only on a vertical grid), "decomp_id" should be specified to *-1*.
  - grid_id [INTEGER; IN]: The ID of the corresponding grid. If the declared argument is a scalar variable, "grid_id" should be set to *-1*. When both "grid_id" and "decomp_id" are not -1, the must correspond to the same component model, and the horizontal grid corresponding to "decomp_id" must be a sub grid of the grid corresponding to "grid_id".
  - dim_size [INTEGER, DIMENSION(:), OPTIONAL; IN]: An array each element of which specifies the size of a dimension of the data buffer pointer. The array size of "dim_size" must be no smaller than the number of dimensions of "data_pointer". "dim_size" should be provided when "data_pointer" points to an empty space (does not point to an existing memory space).
  - field_unit [CHARACTER, OPTIONAL; IN]: The unit of the declared argument. Default unit specified in the configuration file "public_field_attribute.xml" (Please refer to Section 3.2 for details) will be used when "field_unit" is not specified when calling this API.
  - annotation [CHARACTER, OPTIONAL; IN]: The annotation used to mark the corresponding model code that calls this API. It has a maximum length of 512 characters.

- **Description of this API**
  This API declares an argument of the corresponding external module instance, and returns the ID of the new argument when the declaration succeeds. "field_name", "decomp_id", and "grid_id" are keywords of the declared argument. If the "data_pointer" points to an existing memory space, the declared argument will use this memory space, and the size of the memory space must be the same as the required size; Otherwise, "data_pointer" will be assigned to point to a memory space managed by C-Coupler, and such memory space will be shaped following the dimension sizes specified by the "dim_size". All MPI processes of the corresponding component model should call this API for starting the declaration of an argument, while more calls of this API will be required when the argument has multiple chunks in the current MPI process (each call of this API declares a chunk of the argument).





**API 9: CCPL_external_modules_para_get_field_grid_ID**

- **integer FUNCTION CCPL_external_modules_para_get_field_grid_ID (instance_id, field_name, annotation)**
    - return value [INTEGER; OUT]: The grid ID of the corresponding model field instance.
    - instance_id [INTEGER; IN]: The ID of the external module instance.
    - field_name [CHARACTER; IN]: The name of the model field instance. It has a maximum length of 80 characters. Each character must be 'A'~'Z', 'a'~'z', 0~9 or '_'.
    - annotation [CHARACTER, OPTIONAL; IN]: The annotation used to mark the corresponding model code that calls this API, which is recommended but not mandatory, and should be provided by the user. It has a maximum length of 512 characters.

- **Description of this API**
    This API returns the grid ID of the model field instance corresponding to the given field name, which has been registered to C-Coupler in the component model of calling the external module. This API can be called at all stages of an external module.

**API 10: CCPL_external_modules_para_get_field_decomp_ID**

- **integer FUNCTION CCPL_external_modules_para_get_field_decomp_ID (instance_id, field_name, annotation)**
    - return value [INTEGER; OUT]: The parallel decomposition ID of the corresponding model field instance.
    - instance_id [INTEGER; IN]: The ID of the external module instance.
    - field_name [CHARACTER; IN]: The name of the model field instance. It has a maximum length of 80 characters. Each character must be 'A'~'Z', 'a'~'z', 0~9 or '_'.
    - annotation [CHARACTER, OPTIONAL; IN]: The annotation used to mark the corresponding model code that calls this API, which is recommended but not mandatory, and should be provided by the user. It has a maximum length of 512 characters.

- **Description of this API**
    This API returns the parallel decomposition ID of the model field instance corresponding to the given field name, which has been registered to C-Coupler in the host component model of the external module. This API can be called at all stages of an external module.

**API 11: CCPL_external_modules_inst_init**

- **integer FUNCTION CCPL_external_modules_inst_init (host_comp_id, inst_name, modules_name, ptype, dl_name, process_active, control_vars, field_inst_ids, timer_ids, annotation)**
    - return value [INTEGER; OUT]: The ID of the instance of external modules.
    - host_comp_id [INTEGER; IN]: The ID of the host component model that calls the API.
    - inst_name [CHARACTER; IN]: The name also the keyword of the external module instance. It has a maximum length of 80 characters. Each character must be 'A'~'Z', 'a'~'z', 0~9 or '_'.
    - modules_name [CHARACTER; IN]: The name of the external modules, which specifies the information in the corresponding XML configuration file. It has a maximum length of 80 characters. Each character must be 'A'~'Z', 'a'~'z', 0~9 or '_'.
    - ptype [CHARACTER; IN]: The type of the external module, including "individual" and "package". The type "individual" means a unique procedure, while the type "package" means a package that contains multiple procedures specified through the corresponding configuration files. When it is a unique procedure, "modules_name" must be consistent with the corresponding driving subroutines in the dynamically linked library.
    - dl_name [CHARACTER, OPTIONAL; IN]: The name of the dynamic library that contains the unique external module. It should be specified when the value of "ptype" is "individual". It has a maximum length of 80 characters.





Each character must be 'A'~'Z', 'a'~'z', 0~9 or '_'.
- process_active [LOGICAL, OPTIONAL; IN]: A mark for specifying whether current MPI process is active to run and finalize the external module. All processes of the current component model will call the initialization driving subroutine of each external module while only the active processes will call the running driving subroutine of each external module. When "process_active" is not specified, it means that the current process is active.
- control_vars [INTEGER, DIMENSION(:), OPTIONAL; IN]: An integer array of control variables that can be obtained by the external module via the corresponding APIs.
- field_inst_ids [INTEGER, DIMENSION(:), OPTIONAL; IN]: The IDs of model field instances corresponding to the arguments of each external module, e.g., these field instances should cover all input and output arguments of each external module. All these field instances should correspond to the same component model.
- timer_ids [INTEGER, DIMENSION(:), OPTIONAL; IN]: The IDs of timers that can be obtained by the external modules.
- annotation [CHARACTER, OPTIONAL; IN]: The annotation used to mark the corresponding model code that calls this API, which is recommended but not mandatory, and should be provided by the user. It has a maximum length of 512 characters.

- **Description of this API**

This API initializes an external module instance and returns its ID. All MPI processes of the host component model are required to call this API at the same time, with the consistent parameters. This API will load in the corresponding DLLs and call the initialization driving subroutine of the corresponding external modules.

**API 12: CCPL_external_modules_inst_run**

- **SUBROUTINE CCPL_external_modules_inst_run (instance_id, chunk_index, annotation)**
  - instance_id [INTEGER; IN]: The ID of an external module instance.
  - chunk_index [INTEGER; IN]: The index of the given chunk corresponding to single-column data structures. If the corresponding external modules do not use single-column data structures, the chunk index should be -1. The index of the first chunk is 1.
  - annotation [CHARACTER, OPTIONAL; IN]: The annotation used to mark the corresponding model code that calls this API, which is recommended but not mandatory, and should be provided by the user. It has a maximum length of 512 characters.

- **Description of this API**

This API execute an external module instance, where the running driving subroutine of each corresponding external module will be called. All MPI processes of the host component model are required to call this API at the same time, with the same input parameters.

**API 13: CCPL_external_modules_inst_finalize**

- **SUBROUTINE CCPL_external_modules_inst_run (instance_id, annotation)**
  - instance_id [INTEGER; IN]: The ID of an external module instance.
  - annotation [CHARACTER, OPTIONAL; IN]: The annotation used to mark the corresponding model code that calls this API, which is recommended but not mandatory, and should be provided by the user. It has a maximum length of 512 characters.

- **Description of this API**

This API finalize an external module instance, where the finalization driving subroutine of each corresponding external module will be called. All MPI processes of the host component model are required to call this API at the same time, with the same input parameters.



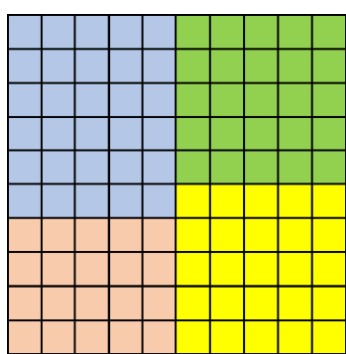

**Figure 1. Example of parallel decomposition on a horizontal grid using four processes to run a component model. Each color represents a subdomain assigned to a distinct process.**





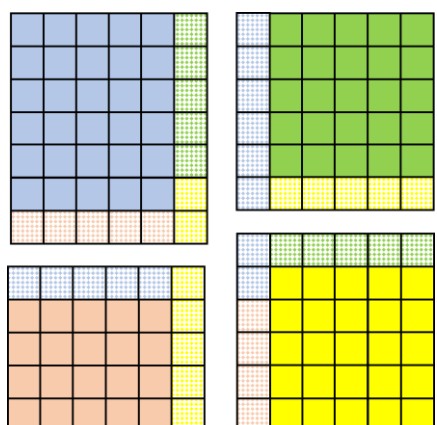

**Figure 2. Example of halo regions corresponding to the parallel decomposition in Fig. 1. Spotted boxes around each subdomain denote each process's halo region; the color of each spotted box indicates the corresponding source process.**





Table 1. The time of coupling initialization and the average memory usage per process of the test coupled model under different global resolutions (cubed-sphere grids) and process numbers for two toy component models. The time taken for parallel remapping weight generation by C-Coupler3.0 is also listed. The time and memory usage of coupling initialization was measured when the remapping weight files had already been generated.

| Resolution of toy comp1 (grid size) | Resolution of toy comp2 (grid size) | Procs of comp1 | Procs of comp2 | Time (s) of coupling initialization | Average memory usage (MB) | Time (s) of remapping weight generation | Size (GB) of remapping weight files |
|---|---|---|---|---|---|---|---|
| 0.0375° (34588806) | 0.075° (8654406) | 1920 | 1920 | 7.92 | 601 | 99.91 | 2.5 |
| | | 3840 | 3840 | 7.27 | 624 | 65.31 | 0.96 |
| | | 7680 | 7680 | 8.50 | 649 | 65.21 | |
| 0.0375° (34588806) | 0.05° (19461606) | 1920 | 1920 | 12.19 | 594 | 127.12 | 3.0 |
| | | 3840 | 3840 | 10.71 | 617 | 86.71 | 2.1 |
| | | 7680 | 7680 | 14.15 | 640 | 93.23 | |
| 0.025° (77803206) | 0.075° (8654406) | 1920 | 1920 | 18.85 | 629 | 179.67 | 5.6 |
| | | 3840 | 3840 | 15.58 | 649 | 150.31 | 1.8 |
| | | 7680 | 7680 | 15.81 | 658 | 130.31 | |
| 0.015° (175024806) | 0.0375° (34588806) | 7680 | 7680 | 40.41 | 683 | 556.89 | 15.0 |
| | | 15360 | 15360 | 56.50 | 729 | 592.44 | 5.9 |
| | | 23040 | 7680 | 56.74 | 791 | 645.56 | |



Table 2. Performance of halo exchange for different bundle settings of fields (five 2-D and five 3-D fields), horizontal grid sizes, and numbers of processes. The 3-D fields include 50 levels. The data type is double floating point (8 bytes). The parallel decomposition is regular (rectangle based) on the horizontal grid, and the halo region on each process has two levels.

| Horizontal grid size | Number of processes | Size (MB) of data transferred by each process | Performance (MB/s) of halo exchange per process | | | |
|---|---|---|---|---|---|---|
| | | | Each field separately | 2-D separately, all 3-D together | All 2-D together, 3-D separately | All fields together |
| 1000×1000 | 20×25 | 0.70 | 14.4 | 54.2 | 57.8 | 109.6 |
| | 40×50 | 0.37 | 8.9 | 24.8 | 22.3 | 66.0 |
| 2000×2000 | 40×50 | 0.71 | 9.3 | 25.1 | 25.5 | 70.3 |
| | 80×100 | 0.38 | 4.9 | 13.4 | 12.8 | 38.7 |
| 4000×4000 | 80×100 | 0.72 | 8.0 | 22.0 | 20.0 | 60.4 |
| | 160×200 | 0.37 | 3.0 | 9.6 | 9.9 | 35.2 |





Table 3. Performance of halo exchange for five 3-D fields (bundled together) under different horizontal grid sizes and numbers of processes. The 3-D fields include 50 levels. The data type is double floating point (8 bytes). The parallel decomposition is regular (rectangle based) on the horizontal grid, and the halo region on each process has four levels.

| Horizontal grid size | Number of processes | Size (MB) of data transferred by each process corresponding to 0% "land" cells | Performance of halo exchange per process under different proportions of "land" cells in halo regions | | | |
|---|---|---|---|---|---|---|
| | | | Performance (MB/s) under 0% useless cells | Speedup under 20% useless cells | Speedup under 50% useless cells | Speedup under 80% useless cells |
| 1000×1000 | 20×25 | 1.42 | 113.7 | 1.13 | 1.76 | 3.21 |
| | 40×50 | 0.79 | 69.5 | 1.10 | 1.70 | 3.01 |
| 2000×2000 | 40×50 | 1.46 | 74.0 | 1.13 | 1.73 | 3.12 |
| | 80×100 | 0.80 | 52.2 | 1.12 | 1.75 | 2.98 |
| 4000×4000 | 80×100 | 1.48 | 58.2 | 1.14 | 1.78 | 3.01 |
| | 160×200 | 0.80 | 38.2 | 1.05 | 1.42 | 2.20 |



Table 4. Performance of data exchange handled by the module-integration framework for an individual DA algorithm (DA1), an ensemble-mean DA algorithm (DA2), and an ensemble-gather DA algorithm (DA3). The overhead of C-Coupler3.0 in initializing a DA algorithm is also shown. Two 2-D and four 3-D fields (50 levels) are assimilated by each DA algorithm. The data type is single floating point (4 bytes).

| Horizontal grid size (resolution) | Number of ensemble members | Size (GB) of the data exchanged between the model ensemble and a DA algorithm | | | Processes for each ensemble instance of component model | Time (s) of initialization | | | Performance (GB/s) of the on-line data exchange | | |
|---|---|---|---|---|---|---|---|---|---|---|---|
| | | DA1 | DA2 | DA3 | | DA1 | DA2 | DA3 | DA1 | DA2 | DA3 |
| 3600×1801 (10km) | 10 | 97.579 | 48.79 | 97.579 | 400 | 3.08 | 20.37 | 19.708 | 728.5 | 109.8 | 110.5 |
| | | | | | 800 | 2.92 | 19.20 | 18.068 | 1566.6 | 233.5 | 210.6 |
| | | | | | 1200 | 2.70 | 18.35 | 17.942 | 2085.0 | 360.5 | 329.9 |
| | | | | | 1600 | 2.73 | 17.70 | 18.589 | 2927.6 | 436.0 | 415.2 |
| | | | | | 2000 | 2.68 | 17.71 | 17.512 | 3519.8 | 650.9 | 557.0 |
| 1800×901 (20km) | 10 | 24.408 | 12.204 | 24.408 | 400 | 0.97 | 9.18 | 7.58 | 741.9 | 130.1 | 123.3 |
| | | | | | 800 | 0.98 | 7.85 | 6.85 | 1492.7 | 258.3 | 226.2 |
| | | | | | 1200 | 0.94 | 9.22 | 7.10 | 2428.2 | 462.5 | 342.5 |
| | | | | | 1600 | 0.97 | 9.25 | 7.13 | 3517.5 | 597.3 | 484.5 |
| | | | | | 2000 | 0.96 | 8.65 | 8.51 | 4614.9 | 784.3 | 601.8 |
| 1800×901 (20km) | 20 | 48.816 | 24.408 | 48.816 | 200 | 0.90 | 25.70 | 22.575 | 777.5 | 123.3 | 114.0 |
| | | | | | 400 | 1.03 | 34.66 | 22.84 | 1481.3 | 215.0 | 205.8 |
| | | | | | 600 | 1.22 | 44.29 | 29.84 | 2159.2 | 387.3 | 260.0 |
| | | | | | 800 | 1.41 | 66.17 | 30.93 | 2894.0 | 566.0 | 359.9 |
| | | | | | 1000 | 1.68 | 88.66 | 35.78 | 3784.8 | 741.4 | 575.0 |

