# Peer review of "C-Coupler3.0: an integrated coupler infrastructure for Earth system modeling"

_Geoscientific Model Development, 2022_

## Author Response (AR1)

We thank the first reviewer for the comments and suggestions. We have modified the manuscript accordingly. In the following, we will reply them one by one.

1. "It would be better to provide many more comparisons with existing couplers used in other models (e.g., GFDL model's coupler, OASIS or CESM's CMEPS). … It would be fairer to compare with other distributed new-generation couplers, in terms of performance, usability and accuracy (and property preservation, e.g., during remapping and grid triangulation, see Ullrich and Taylor, 2015). …"

**Response**: For revision, we added the comparison about coupling initialization time between OASIS3-MCT5.0 and C-Coupler3.0. The results show that the both couplers can achieve fast coupling initialization and C-Coupler3.0 is faster. We did not evaluate the accuracy of remapping because the development of C-Coupler3.0 did not include the improvement about remapping algorithms. We will try to improve the accuracy of with high-order remapping algorithms in the future. Please refer to P13L11~L14, Table 2 (P29), and P16L7 of the revised manuscript.

2. Common halo-exchange library: please clarify what is already in v2.0 and what is new in v3.0. It is only the checksums APIs and global halo indexing (instead of the special value labels) that are new?

**Response:** Yes. Only the APIs CCPL_register_halo_region, CCPL_register_halo_exchange_interface, CCPL_execute_halo_exchange_using_id, CCPL_finish_halo_exchange_using_id, CCPL_report_one_field_instance_checksum and CCPL_report_all_field_instances_checksum are new. Please refer to P8L11~L14 of the revised manuscript.

3. "… it should be much faster than the I/O on which the DA data exchange in an offline DA system depends" This point could be modified and clarified.

**Response:** The statement has been modified. Please refer to P15L14~L19 of the revised manuscript.

4. P3, L31: point 2 and point 3 seem to be very similar

**Response:** These two points have been merged into one. Please refer to P3L31~L32 of the revised manuscript.

We thank the second reviewer for the comments and suggestions. We have modified the manuscript accordingly. In the following, we will reply them one by one.

1. Each grid has 3 distributions (index, domain, parallel-decomposition). The native model decomposition is the parallel-decomposition in each component. I assume the 3 distributions for each grid only exist in the coupler component and they are all stored there and that the domain decomposition is mostly used to generate mapping weights files which is done at iniitialization? Why save all 3 distributions in the coupler, why not just use the index distribution in the coupler permanently and allow the other 2 to be temporary?

**Response**: Yes. The domain decomposition is used to generate mapping weights files, and the index distribution is used in the coupler permanently and the other 2 are temporary. This point has been clearly stated in the revised manuscript (P5L12~L14).

2. Does the halo exchange implementation impose any constraints on the width of the halo? If not, maybe mention that explicitly in secton 3 under advantages on page 8.

**Response**: The halo exchange implementation does not impose any constraints on the width of the halo and supports any width of the halo region. Please refer to P8L19 of the revised manuscript.

3. I assume the checksum (page 8) ultimately requires an MPI global reduction, so this is primarily for testing and debugging?

**Response**: Yes. The checksum functionality uses MPI global reduction and it is primarily for testing and debugging. Please refer to P8L27 of the revised manuscript.

4. The module-integration framework description is not very clear. Ultimately, I had a look at the atm_ocn_flux_ccpl code. … You could also just have a layer in-between that translates data types without all the framework and DLL stuff. What extra benefits does the framework and DLL provide and is the extra layer(s) really better than just having a copy in/out at the top of the subroutine being called?

**Response**: The example of the air–sea flux algorithm is used to the module-integration framework can work as an extra layer for using an algorithm. The benefit

of the module-integration framework was further introduced with the DA framework DAFCC1. Please refer to P14L16 and P14L28~P15L3 of the revised manuscript.

5. Finally, the flux_atmocn APIs don't seem to match what is documented in the appendix. Why?

**Response**: We carefully compared the APIs used in integrating the air–sea flux algorithm, which are the same as the APIs documented in the appendix. The misunderstanding may arise from the subroutine interfaces (e.g. *use_ccpl_do_flux_atmocn_init/ use_ccpl_do_flux_atmocn_run*), which calls the APIs of the module-integration framework (e.g. *CCPL_external_modules_inst_init/ CCPL_external_modules_inst_run*) to drive the air–sea flux algorithm.

6. Page 9, line 9. The next few lines seem to discuss subroutines that are called on columns of data. That is really just a minor class of types of methods that might be called. I don't think that restriction is necessarily related to the module-integration framework. You might be trying to say the module-integration framework is more flexible than CCPP. If so, say that, NOT page 9 lines 9-16.

**Response**: We clearly stated that the module-integration framework is aimed to be more common than CCPP. Please refer to P9L29.

7. In table 1, is the coupling initialization time including reading mapping weights? The remapping weight generation time is done with a separate test?

**Response**: Yes. It has been clearly stated at the title of Table 1 that the time and memory usage of coupling initialization was measured when using offline remapping weight files that had already been generated. Please refer to P28.

8. In Table 2 and 3 captions (and section 7.2), you say "the halo region on each process has two/four levels". Please clarify what "levels" means. Is that the width of the halo?

**Response**: Yes. "levels" means width of the halo. Please refer to the Table 3 and 4 captions (P30 and P31).

9. The results are presented primarily as MB or GB per second. Can you provide an idea of the typical overhead associated with coupling for some working systems

for C-Coupler3? Is it 0.1%, 1%, 5%, 20%? How does that compare to a similar application using C-Coupler2?

**Response**: We did not evaluate the overhead of C-Coupler3.0 in the simulation run of a real coupled model system because C-Coupler3.0 accelerates coupling initialization but does not change the codes of data transfer and data interpolation in C-Coupler2. Moreover, the overhead of coupler in a coupled system also depends on numerical algorithms and parallel implementations of each component model. We did not evaluate the coupling initialization time of C-Coupler2 under the test cases in Table 1 because C-Coupler2 runs out of memory under each case. Please refer to P13L19~L21 of the revised manuscript.

---

## Author Response (AR2)

Dear Editors,

Thanks a lot for the remarks. In the following, we will reply them one by one.

1. Your reference list includes works "in review". Such works can be cited upon submission if being available to the reviewers. They should not be cited in the final, accepted manuscript, unless published, accepted for publication, or available as preprint with a DOI.

Response: We have modified the reference "Shi et al., 2021" into "Shi et al., 2022" that was finally published by GMD. The other two preprints referenced (Heinzeller et al., 2022; Yu et al., 2022) are still kept as they have DOI.

2. With the next revision, please re-label your figures in the Supplement to "Figure S1", "Figure S2" etc..and your tables to "Table S1", "Table S2" etc. ad adjust the references in the main text accordingly.

Response: We did not change the labels of the Figures and Tables in the Supplement because no figure or table is directly referenced in the main text.